# Flow Dynamic Pattern in Liver and Renal Transplantation under Exercise Prescription Program

**DOI:** 10.3390/jcm12134521

**Published:** 2023-07-06

**Authors:** Marco Corsi, Edoardo Falconi, Roberto Palazzo, Vittorio Bini, Gabriele Mascherini, Sabrina Mancini, Marco Maglione, Laura Stefani

**Affiliations:** 1Sports Medicine Center, Clinical and Experimental Department, University of Florence, 50121 Florence, Italy; marco.corsi@unifi.it (M.C.); edoardo.falconi@unifi.it (E.F.); roberto.palazzo@unifi.it (R.P.); 2Department of Medicine, University of Perugia, 06123 Perugia, Italy; binivittorio@gmail.com; 3Clinical and Experimental Department, University of Florence, 50121 Florence, Italy; gabriele.mascherini@unifi.it; 4Clinical and Experimental Department, School of Human Health Science, University of Florence, 50121 Florence, Italy; sabrina.mancini@unifi.it; 5CV Ultrasound Division, ESAOTE Spa, 50127 Florence, Italy; marco.maglione@esaote.com

**Keywords:** HyperDoppler, organ transplantation, diastolic function, exercise prescription

## Abstract

Background: Cardiovascular diseases in the context of renal and liver transplants remain the leading cause of morbidity and mortality. Physical exercise at a moderate intensity is allowed to contrast the risk profile. Echocardiographic evaluation is essential to stratifying potential cardiotoxicity by the standard and, more recently, the deformation and dynamic study of the intracardiac vortex. This study aims to investigate the vortex echo parameters of solid-organ-transplanted subjects who are physically active compared to a control group of healthy subjects. Methods: A group of 33 transplanted subjects (16 kidneys and 17 livers) was studied via a transthoracic echocardiography exam, comprehending the myocardial deformation parameters of global longitudinal strain (GLS), twisting of the left ventricle (LV) chamber, and HyperDoppler image acquisition. Results: The subjects enrolled in this study were 50 in total: there were 33 transplanted and 17 healthy subjects. The transplanted subjects presented higher values of interventricular septum in diastole (IVSd *p* = 0.001), posterior wall diastolic (PWd *p* = 0.05), and left ventricle mass index (LVMI *p* = 0.029); ejection fraction (EF) was found to be higher in athletes (*p* < 0.001). Transplanted subjects presented mild diastolic dysfunction, emerging only from septal E values (*p* = 0.001). The 4DStrain (*p* = 0.018) and GLS2c (*p* = 0.017) were significantly better in the athletes. All of the geometrical and energetical vortex data were in the normal range and no significant differences were found. An interesting positive correlation was evident for the diastolic parameter, particularly the E/A ratio (*p* = 0.023) and E’ septal value (*p* = 0.049), along with the vorticity fluctuation. This behavior was present for all subjects, particularly those that were transplanted (*p* = 0.005). Conclusions: In the vortex investigation, especially in cases of normal EF, the positive correlation of some diastolic parameters with the flow dynamic patterns corroborates this hypothesis. The HyperDoppler analysis could be helpful to detecting potential damage earlier in the diastolic time before a systolic deficiency.

## 1. Introduction

Solid-organ-transplanted subjects represent a new category of frail patients, potentially appropriate for several physically tailored exercise programs [1]. Despite the surgical treatment, they maintain a high cardiovascular risk of traditional and specific cardiovascular risk factors [2,3]. Studies have demonstrated that regular moderate-intensity training has a cardio-protection role regarding many diseases and transplanted populations [4], reducing all causes of mortality and cardiovascular diseases. Specific positive effects are described mainly in renal-transplant cases [5]. A healthy lifestyle and physical activity are recommended to improve morbidity and cardiovascular outcomes [6]. Liver-transplanted subjects are less studied; however, specific literature reports that lifestyle intervention involving physical activity reduces cardiovascular risk factors [7,8] via an outdoor or home-based regimen. Correct cardiological screening is therefore important in asymptomatic transplanted subjects [9] during the management of exercise intensity, especially in detecting any potential acute event as a consequence of potential cardiotoxicity and a previous prolonged sedentarism. The echocardiographic parameters are well known to establish the degree of systolic and diastolic dysfunction [10]. In addition to the strain analysis primarily studied in terms of several chronic conditions, and also in terms of transplantation regarding systolic time [11], more recently, the new parameters relating to the intracardiac fluxes [12,13] are emerging tools designed to better identify the myocardial function in healthy individuals and in patients [14]. During the cardiac cycle, the vortex flow changes and during the isovolumic contraction phase, blood is redirected toward the LV outflow tract [15]. Studies have shown that two vortices can be recognized within the LV during diastole. The concept is based on the intraventricular vortex possibly contributing to a higher flow of energetics expenditure. The vortex pattern in liver- and kidney-transplanted subjects has yet to be studied. The principal aim of this study was to compare the echocardiographic parameters of the transplanted subjects, who were regularly trained at moderate intensity, with a healthy group and verify vortex analysis’ potential contribution to detecting a major risk profile. 

## 2. Materials 

Adult subjects with a mean age of 55 who regularly trained at moderate intensity that were referred to the sports medicine unit in 2022 after solid-organ transplantation, for at least 1 year, were enrolled in this study. The inclusion criteria were that participants had to have been transplanted for at least one year and clinically stable in the absence of acute cardiovascular symptoms and arrhythmic events in the few months before this study. In addition, a reasonably good acoustic window, suitable for acquiring specific echocardiographic parameters, was fundamental among the inclusion criteria. Comorbidities, such as diabetes, arterial hypertension, or other metabolic diseases, were not, on the contrary, a reason for exclusion. In case of hypertension, they were treated using antihypertensive treatments (calcium-channel blockers, alpha-blockers, ACE inhibitors, or ARBs) and immunosuppressive therapy, including drugs such as calcineurin inhibitors (Ciclosporin or Tacrolimus), in combination with Mycophenolate or Everolimus and steroids (Methylprednisolone). None of them received beta-blockers. As is normal, participants were screened in the ambulatorial setting and before starting a physical exercise program at a sports medicine center, the biochemical parameters were obtained. Only the subjects with normal values were included in this study. 

The causes of liver transplantation were mainly due to previous HCV liver cirrhosis or, in other cases, sclerosing cholangitis or primary biliary cirrhosis. In the kidney transplantation group, the leading causes were polycystosis, pyelonephritis, post-glomerulonephritis, and Berger’s nephropathy.

The participants were all included in a tailored physical exercise prescription program that they followed for at least 1 year before the investigation. The program consisted of moderate-intensity activity, individually established using the Karvonen formula and, if necessary, modified in the case of beta-blocking drugs [16]. The participants’ heart rates during the aerobic exercise were obtained by the cardiopulmonary test and fixed at one threshold level. The physical activity prescribed was mixed (aerobic and counter-resistance) thrice weekly. At a moderate intensity of about 60% of the maximal heart rate, aerobic exercise was suggested to be undertaken for at least 30 min. The aerobics session was followed by resistance training, including exercises involving at least eight groups of body muscles. All of the echocardiographic measurements were obtained in rest conditions. Data were interpreted following echocardiography guidelines [17]. The deformation parameters are used mainly in many metabolic chronic diseases and have recently been indicated for use for transplanted subjects. The dynamic flow pattern determined by HyperDoppler analysis was also obtained for this investigation [10].

The control group consisted of active subjects, similar in age, who were periodically evaluated in the ambulating setting of the sports medicine unit. They were regularly trained following a program of mixed exercise at a moderate intensity that was previously established by specific exercise texts. The personalized exercises were prescribed at least thrice weekly for at least 30 min per session.

## 3. Methods 

### 3.1. Echocardiographic Evaluation 

Myocardial function was assessed by a transthoracic echocardiography exam using the MyLABX8 exp Esaote echocardiograph equipped with a PX1-5-2.5 MHz probe. A certified cardiologist was involved in the acquisition of the images.

The morpho-functional and Doppler system-diastolic parameters were considered according to the ASE/ESC 2015 guidelines [18]. The following standard parameters were considered: diameter and thickness of the left ventricle, ejection fraction (EF) by the SR (Simpson rule) method, cardiac mass indexed (CMI), left atrial volume (LAV), and pulmonary pressure (PP). Regarding the diastolic function, the mitral inflow pattern (wave E/A ratio, deceleration time), annular mitral septal, and lateral TDI with E/e ratio were considered. Diastolic function was classified as normal or abnormal with impaired relaxation (grade 1), pseudo-normal (grade 2), or restrictive (grade 3). In the case of valvular heart disease, the severity of the valve dysfunction was evaluated according to the ASE 2015 guidelines [17]. In addition, 1-cycle clip images from 4C, 2C, 3C, and parasternal short-axis projections on the mitral and apical planes were acquired to evaluate strain, rotation, and twisting. From the 3C chamber projection, with Color Doppler with a maximum FR of 21 HZ, clips were acquired in at least three cycles to calculate the intracardiac vortices via HyperDoppler software. The relative geometric and energetic parameters were obtained.

### 3.2. Strain Analysis

All subjects, transplanted and controls, completed the echocardiographic examination for the evaluation of myocardial deformation parameters, including the global longitudinal strain (GLS) (2 chambers, 3 chambers, 4 chambers), 4D strain, and calculation of the twisting of the left ventricular chamber obtained by the difference of the basal rotation compared to the apical one. A dedicated software (Xstrain tm-ESAOTE-Genoa, Italy) was used to calculate the myocardial deformation. According to the criteria established in the EACVI/ASE consensus document, it was possible to obtain the results of the overall study and the average myocardial deformation by acquiring the images over the entire cycle at a high frame rate. In particular, the diastolic index of early myocardial deformation, the early strain rate (eSR), was also calculated and subsequently expressed as the E/e’ strain rate ratio. The E1/eSR ratio was considered as a tool to evaluate the diastolic dysfunction grade, as reported in the literature [10].

### 3.3. HyperDoppler Image Acquisition and Measurements

HyperDoppler image acquisition was performed using the same ESAOTE MyLab X8 exp echo scanner and PX1-5 cardiac probe (ESAOTE, Florence, Italy) used for the conventional examination, with subjects in the left lateral decubitus position, as previously described [19,20].

A standard apical 3C view that included the aortic tract was used. Particular attention was paid to including the LV cavity and the LV outflow tract (LVOT) within the corner of the color Doppler sector as much as possible. The depth and width of the sector were set to achieve a Color Doppler frame rate of 21 fps. The repetition frequency of the Color Doppler pulse was 4.4 MH.

For the evaluation of the intracardiac vortices, the acquisition of 3 cardiac cycles, in cine-loop three-color cameras that were no lower than 21 Hz, was used with subjects in the left lateral decubitus position using HyperDoppler software included in the MyLab X8 exp. The Color Doppler long-axis view was acquired at the end of the expiration in a cine-loop format, including two consecutive cardiac cycles, and was stored in the echo scanner for analysis (Figure 1).

### 3.4. Statistical Analysis

The Mann–Whitney test was utilized to compare continuous or discrete variables between groups; whereas, the correlations were tested using the Spearman rho coefficient. All statistical analyses were performed using IBM-SPSS^®^ version 26.0 (IBM Corp., Armonk, NY, USA, 2019) and a two-sided *p*-value ≤ 0.05 was considered significant.

## 4. Results

The number of transplanted subjects enrolled in this study was 33 and they were within the required range of time regarding transplantation, from 1 year up to 10 years. The group was composed of sixteen kidney transplants (nine men and seven women) and seventeen liver transplants (thirteen men and four women), with their mean age being 55 years. No differences have been found within the transplanted group in terms of gender number (*p* = 0.388). The comparison of the two groups for the specific parameters did not show any significant differences. The BMI parameter was normal in the transplanted participants (24.7 ± 2.9 with *p* = 0.066 if compared to controls), as heart rate (HR: 72 ± 1 bpm), SBP and DBP (respectively 126.3 ± 3 mmHg; 78.3 ± 6.4 mmHg), which were in the normal range and were not different to the controls. Particularly in the presence of concomitant disease, the transplanted group was homogeneous: diabetes was present in 14 liver-transplanted subjects and in 10 kidney-transplanted subjects. Hypertension, controlled by pharmacological treatments, was evident in 13 liver transplants and in 10 kidney transplants. 

The general data and echocardiographic parameters of the analyzed population are displayed in Table 1. All values expressed as means and standard deviations were normal despite being significantly different compared to the controls. Statistical differences were found in the LV wall thickness in the LVMI, with major values for the transplanted subjects; meanwhile, the EF values were higher in athletes. Despite being clinically asymptomatic, the transplanted subjects presented a mild diastolic dysfunction, emerging only from septal E’ value analysis and in normal systolic functioning (Table 2).

No differences were found in the rotation and twist parameters of the LV chamber; meanwhile, the 4DStrain and GLS2c, as expressions of myocardial performance, were significantly better in athletes.

All of the geometrical and energetical vortex data were in the normal range. The data agreed with the results recently reported in the literature [15]. No significant differences were found in transplanted subjects when compared to the controls, as shown in Table 3. An interesting positive correlation was evident for the diastolic parameter, E/A ratio, and E’ septal value regarding the vorticity fluctuation. This behavior is present in all cases and particularly for those with transplants. This could be referred to as a more vortices flow with significant energy dissipation.

Regarding the correlation analysis, the whole group showed a correlation of the E/A with vorticity fluctuation (*p* = 0.023; rho = −0.341) and a vorticity parameter with septal E’ (*p* = 0.049; rho = 0.299). The control group showed a correlation of septal E’ with shear stress fluctuation (*p* = 0.019; rho = −0.661), E/e’ with vorticity fluctuation (*p* = 0.026; rho = −0.636), and septal A’ with vortex area (*p* = 0.037; rho = −0.605) and kinetic energy fluctuation (*p* = 0.049; rho = −0.577). In the transplanted group, E/A correlated with vorticity fluctuation (*p* = 0.005; rho = −0.483) (Table 4).

## 5. Discussion

The long-term exposure to prolonged physical inactivity and the potential cardiotoxicity, due to the common assumption of immune-suppressive drugs and the ongoing inflammation process, is responsible for the “post-transplant syndrome” that maintains high cardiovascular risk factors. The echocardiographic evaluation is a determinant in the first screening of the subjects involved in specific exercise programs and the follow-up [7,9]. These results are significant considering that the liver-transplanted category still needs to be well investigated [21]. These results are significant considering that the liver-transplanted category needs to be well investigated and benefits regarding the cardiovascular profile remain uncertain. 

The new and emerging echocardiographic parameters could help detect potential cardiac damage due to prolonged immunosuppressive therapy. The effective compartmentalization of the hydration exposes patients to heart failure, often not associated with a specific systolic dysfunction. Recent experiences involving vortex analysis have shown that the flow pattern can change due to myocardial remodeling, with evidence of differences in some cases [15]. Authors have demonstrated that DCM has a different vortex pattern when compared to athletes or normal subjects. In our study, the category investigated is not includible in the heart failure category; however, despite preserved EF, the diastolic pattern is compatible with mild dysfunction. The results show a specific correlation between vortex parameters and diastolic values, even if the vortex data are not significantly different compared to the control group. Previous studies support the importance of having a predominant systolic dysfunction to determining a vortex modification. The main difference between DCM and athletes is evident [15]; on the contrary, the data obtained in the present study, based on the correlation found, could suggest the potential role of these parameters in cases of progressive diastolic dysfunction; they may be higher than that of the group studied. The correlation between at least two vortex parameters (Vorticity Fluctuation and E/A and septal E1) and a vortex area with an A1 septal value confirms the potential role of the vortex parameters within the diastolic phase. It could be reasonable to hypothesize that in the presence of higher diastolic dysfunction expressed by an E/E1 higher ratio value, the vortex parameters show a peculiar trend characterized by a reduced vortex area and higher vorticity fluctuation, as reported in Table 3, despite not being significantly modified. In this context, the dynamic flow evaluation might have a specific role in defining the diastolic function and the correct management of the subjects under the exercise prescription treatment.

The subtle diastolic dysfunction is not evident from the traditional 2D echo data. It could be better investigated by some vortex parameters that are strictly related to a vortices flow that is inadequate. This could offer some opportunities for an extended exercise program.

## 6. Conclusions

This pilot study describes the method’s feasibility in this category and gives more significance to its potential role in defining a critical condition regarding these subjects. For the subjects, the prognostic outcome represents the need to prioritize and tailor their physical activity regimen [9]. Although the solid-organ transplantation group in this context is mixed and composed of liver and kidney transplants, the population seems to be homogeneous in terms of general characteristics and these preliminary results highlight the potential contribution of the new echo parameters in cardiological evaluation, especially if the data are within the normal range. For this reason, healthy subjects have been chosen as the control group here, in agreement with a previous study where the vortex analysis involved normal subjects and athlete subjects [15]. The data found agree with previous studies describing the vortex values by age [22]. The clinical significance of the vortex analysis and the close links between the diastolic markers are in agreement with previous studies and are also hypothesized in this category [22,23,24]. 

In any case, despite the details of the single parameter related to the geometrical characteristics of the cardiac flow pattern, the vortex profile could be considered and proposed as a new immediate analysis of the flow dynamicity of the heart and potentially of the effects of physical exercise on the subjects. The intracardiac direction along the major axis of the LV chamber can support the other standard measurement of the LV chamber. More studies will be necessary to validate the specific role of these parameters in specific categories and levels of myocardial system-diastolic dysfunction. 

## 7. Limitations

Although this study may be considered original in the context of post-transplanted subjects, some limitations can be found. Among them, the first point is the low number of cases investigated. However, it is to be considered that:

This study sample includes a very special population (subjects with organ transplants with preserved systolic function);Subjects were investigated in a single-center study;A single expert sonographer took measurements.

These three elements limit variability; however, they improve statistical power.

## Figures and Tables

**Figure 1 jcm-12-04521-f001:**
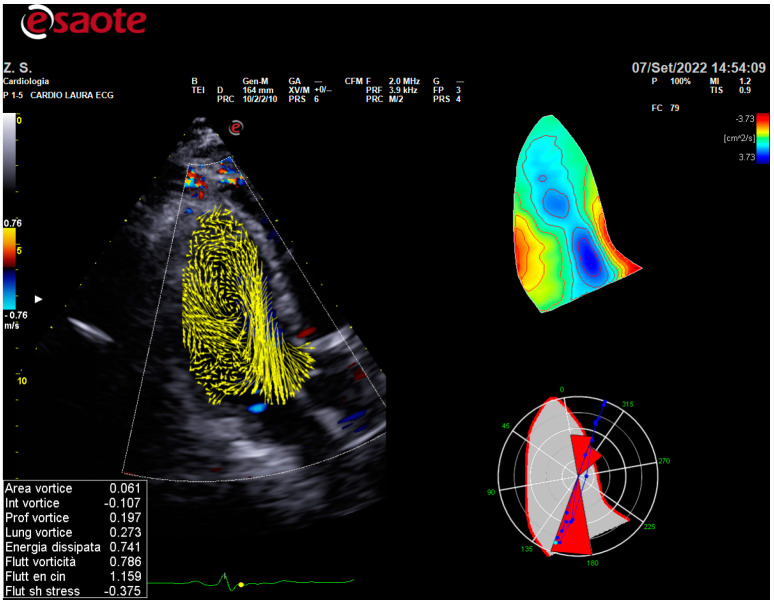
HyperDoppler. Example of reconstructions from 3C view using HyperDoppler analysis.

**Table 1 jcm-12-04521-t001:** General and echocardiographic parameters.

	All Subjects (n = 50)		All Subjects (n = 50)
** *Systolic Parameters* **		** *Vortex Parameters* **	
IVSd (mm)	9.6 ± 1.2	Vortex Area	0.21 ± 0.07
PWd (mm)	9.3 ± 1.1	Vortex Depth	0.33 ± 0.11
LVDD (mm)	49.7 ± 4.7	Vortex Length(mm)	0.47 ± 0.26
LVSD (mm)	30.6 ± 4.2	Vortex Intensity	−0.28 ± 0.22
LVMI (g/m^2^)	96.1 ± 20.8	Energetic Dissipation	0.77 ± 0.35
RWT (cm)	0.37 ± 0.05	Kinetic Energy Fluctuation	0.95 ± 0.15
EF (%)	62.9 ± 5.7	Vorticity Fluctuation	0.78 ± 0.13
MAPSE (mm)	17.6 ± 2.2	Shear Stress Fluctuation	−0.39 ± 0.28
TAPSE (mm)	23.7 ± 4	** *Strain Parameters* **	
Aortic Root (mm)	26.7 ± 4.3	4D Global Strain (%)	−16.4 ± 2.5
Aortic Arch (mm)	26.9 ± 3.6	GLS4c (%)	−17.5 ± 3.3
LAVI (mL/m^2^)	24.5 ± 7.7	GLS3c (%)	−15.6 ± 3.1
** *Diastolic Parameters* **		GLS2c (%)	−16.3 ± 3.8
E/A	1.1 ± 0.4	Apical Rotationº	5.5 ± 3.5
DTc (ms)	212 ± 47.9	Basal Rotationº	−4.7 ± 2.5
IVRT (ms)	85.6 ± 18.2	Twistº	8.1 ± 3.7
E/e’	8.3 ± 3.9		

Data are expressed as mean ± standard deviation. IVSd (Interventricular septum in diastole), PWd (posterior wall diastole), LVSD (ventricular diameter in systole), LVDD (ventricular diameter in diastole), LVMI (left ventricular mass index), RWT (relative wall thickness), EF (ejection fraction), MAPSE (excursion of the mitral annular plane), TAPSE (excursion of the tricuspid annular plane), LAVI (indexed left atrial volume), DTc (deceleration time), E/A (ratio between E and A peak), DTc (deceleration time), IVRT (isovolumetric relaxation time), E/e’ ratio (ratio between E peak wave and septal E’). GLS: global longitudinal strain; 2C: two chambers; 4C: four Chambers; 3C: three chambers.

**Table 2 jcm-12-04521-t002:** Echocardiographic parameters.

	Transplanted (n = 33)	Controls (n = 17)	*p*
** *Systolic parameters* **			
IVSd (mm)	9.9 ± 1.2	8.8 ± 0.8	**0.004**
PWd (mm)	9.5 ± 1.2	8.8 ± 0.9	**0.05**
LVDD (mm)	50.5 ± 4.7	47.9 ± 4.3	0.06
LVSD (mm)	30.9 ± 4.2	29.7 ± 4.4	0.46
LVMI (g/m^2^)	100 ± 21.4	86.1 ± 16	**0.029**
RWT (cm)	0.38 ± 0.05	0.37 ± 0.04	0.32
EF (%)	60.9 ± 4.8	67.5 ± 4.9	**<0.0001**
MAPSE (mm)	17.6 ± 2.2	17.8 ± 2.4	0.84
TAPSE (mm)	23.6 ± 3.7	24.1 ± 4.9	0.58
Aortic Root (mm)	26.6 ± 4.7	26.8 ± 2.9	0.86
Aortic Arch (mm)	27.8 ± 3.6	24.8 ± 2.8	**0.031**
LAVI (mL/m^2^)	25.2 ± 7.7	22.8 ± 7.2	0.23
** *Diastolic Parameters* **			
E/A	0.99 ± 0.34	1.19 ± 0.38	0.11
DTc (ms)	217 ± 52	198 ± 32	0.28
IVRT	83.4 ± 17	91.5 ± 20	0.18
Septal E’	−0.78 ± 0.25	−1.01 ± 3.14	**0.001**
E/e’	9.5 ± 3.6	5.1 ± 2.7	**<0.0001**

Data are expressed as mean ± standard deviation. Boldface = *p* < 0.05. IVSd (Interventricular septum in diastole), PWd (posterior wall diastole), LVSD (ventricular diameter in systole), LVDD (ventricular diameter in diastole), LVMI (left ventricular mass index), RWT (relative wall thickness), EF (ejection fraction), MAPSE (excursion of the mitral annular plane), TAPSE (excursion of the tricuspid annular plane), LAVI (indexed left atrial volume), E/A (ratio between E and A peak), DTc (deceleration time), IVRT (isovolumetric relaxation time), E/e’ ratio (ratio between E peak wave and septal E’).

**Table 3 jcm-12-04521-t003:** Comparison of vortex data between transplanted and controls.

	Transplanted (n = 33)	Controls (n = 17)	*p*
** *Vortex Parameters* **			
Vortex Area	0.198 ± 0.059	0.237 ± 0.083	0.14
Vortex Depth	0.309 ± 0.083	0.367 ± 0.155	0.57
Vortex Length	0.546 ± 0.117	0.293 ± 0.394	0.07
Vortex Intensity	−0.221 ± 0.185	−0.405 ± 0.237	0.07
Energetic Dissipation	0.794 ± 0.396	0.716 ± 0.229	0.76
Kinetic Energy Fluctuation	0.962 ± 0.116	0.914 ± 0.199	0.72
Vorticity Fluctuation	0.810 ± 0.076	0.714 ± 0.199	0.32
Shear Stress Fluctuation	−0.380 ± 0.321	−0.415 ± 0.136	0.66
** *Strain Parameters* **			
4D Global Strain (%)	−15.8 ± 2.2	−18.2 ± 2.8	0.018
GLS4c (%)	−17 ± 3.3	−18.6 ± 3.1	0.15
GLS3c (%)	−15.1 ± 3.2	−17.2 ± 2.5	0.09
GLS2c (%)	−15.4 ± 3.6	−18.9 ± 3.4	0.017
Apical Rotationº	5.07 ± 3.3	7.07 ± 4.1	0.19
Basal Rotationº	−4.73 ± 2.7	−4.41 ± 1.4	0.97
Twistº	7.67 ± 3.6	9.67 ± 3.8	0.11

**Table 4 jcm-12-04521-t004:** Significance of Spearman Correlation between vortex and diastole parameters.

		E	A	E/A	IVRT	DTc	Septal E’	E/e’	Septal A’	Septal S’
Vortex Area	All Subjects	0.34	0.79	0.57	0.83	0.30	0.93	0.99	0.58	0.71
	Transplanted	0.72	0.88	0.88	0.82	0.44	0.87	0.57	0.57	0.79
	Controls	0.54	0.19	0.09	0.56	0.95	0.32	0.62	0.04	0.63
Vortex Depth	All Subjects	0.78	0.87	0.64	0.48	0.47	0.84	0.81	0.28	0.58
	Transplanted	0.50	0.73	0.40	0.77	0.27	0.64	0.87	0.94	0.81
	Controls	0.77	0.60	0.68	0.53	0.93	0.85	0.6	0.08	0.44
Vortex Length	All Subjects	0.42	0.64	0.22	0.80	0.46	0.48	0.63	0.12	0.81
	Transplanted	0.44	0.84	0.85	0.49	0.19	0.51	0.38	0.14	0.89
	Controls	0.13	0.46	0.33	0.51	0.83	0.33	0.57	0.76	0.93
Vortex Intensity	All Subjects	0.22	0.81	0.57	0.88	0.78	0.66	0.51	0.54	0.91
	Transplanted	0.38	0.42	0.37	0.69	0.87	0.93	0.99	0.66	0.57
	Controls	0.87	0.97	0.95	0.91	0.73	0.89	1.0	0.55	0.68
Energy Dissipation	All Subjects	0.54	0.65	0.88	0.54	0.36	0.94	0.77	0.75	0.93
	Transplanted	0.29	0.94	0.68	0.51	0.56	0.76	0.23	0.64	0.49
	Controls	0.46	0.69	0.81	0.89	0.19	0.39	0.06	0.27	0.19
Vorticity Fluctuation	All Subjects	0.27	0.22	0.023	0.48	0.11	0.049	0.29	0.14	0.48
	Transplanted	0.09	0.19	0.005	0.57	0.07	0.11	0.88	0.06	0.93
	Controls	0.93	0.97	0.67	0.03	1	0.20	0.03	0.95	0.46
Kinetic Energy Fluctuation	All Subjects	0.77	0.54	0.66	0.76	0.64	0.49	0.69	0.11	0.84
	Transplanted	0.85	0.47	0.72	0.78	0.62	0.36	0.43	0.67	0.91
	Controls	0.45	0.82	0.95	0.72	0.93	0.96	0.35	0.05	0.89
Shear Stress Fluctuation	All Subjects	0.44	0.45	0.26	0.21	0.71	0.53	0.91	0.06	0.64
	Transplanted	0.67	0.97	0.87	0.18	0.93	0.73	0.65	0.10	0.10
	Controls	0.73	0.07	0.07	0.92	0.79	0.02	0.26	0.24	0.17

## Data Availability

Data can be obtained from Laura Stefani on a reasonable request at laura.stefani@unifi.it.

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
