# Peer review of "Flow Dynamic Pattern in Liver and Renal Transplantation under Exercise Prescription Program"

_jcm, 2023, doi:10.3390/jcm12134521_

Round 1

Reviewer 1 Report

I am grateful for the opportunity to review the manuscript of Corsi et al. "Flow dynamic pattern in liver and renal transplantation". In this article, the authors studied various indicators of echocardiography in patients after liver or kidney transplantation and in healthy individuals during physical training in a sports medicine center. At the same time, the vortex analysis was carried out using HyperDoppler software. The authors obtained, in general, the expected results (in patients, some ECHOCG parameters (EF, LV myocardial mass, parameters of diastolic function) were worse), but not the vortex analysis data.

In addition, when reviewing, I had questions and comments to which I would like to receive answers from the authors of the article.

1. The title and purpose of the article does not match the content (it is not indicated that individuals were evaluated during physical training programs).

2. Characteristics of the surveyed is presented insufficiently. For example, age is listed as 55+/- without DM data. In addition, it is necessary to compare data in groups (gender, age, BMI, presence of concomitant diseases, received therapy, heart rate, SBP and DBP). These parameters can have a significant impact on the ECHOCG parameters and on the revealed differences between the groups.

3. I think table 1 is redundant, if necessary, this information can be presented as an additional column in table 2. However, given the different nature of the groups (patients after transplantation and healthy controls), this is hardly appropriate.

4. It is not clear how loads were selected in patients receiving beta-blockers and how many such patients were in the study?

5. In the group after transplantation, there is no information about the terms after the operation and the main biochemical markers.

6. Reading lines 195-197 ("The range of the vortex parameters has been in investigators and DCM in a study demonstrating significant differences among them, suggesting a potential role in the definition of the single group investigators"), one gets the impression that source link missing.

7. The Discussion section is poorly written - there is practically no comparison of the obtained data with previously published works, it should be revised.

8. I question the authors' assertion at lines 209-211 ("In addition, it could be helpful to use the HyperDoppler analysis to early detect potential damage in the diastolic time before having a systolic impairment") because differences in diastolic function scores between groups were , but not according to HyperDoppler analysis.

9. The numbering of subsections of the manuscript is somewhat chaotic, it needs to be clarified.

10. The article does not contain a Conclusion section.

No comments

Author Response

In first line we  want to thank  the reviewers  for giving us the opportunity to improve  the paper  and for  the  attention  to the  data  obtained. All the suggestions have been considered  and the modifications follow  the  recommendations .

REV1

I am grateful for the opportunity to review the manuscript of Corsi et al. "Flow dynamic pattern in liver and renal transplantation". In this article, the authors studied various indicators of echocardiography in patients after liver or kidney transplantation and in healthy individuals during physical training in a sports medicine center. At the same time, the vortex analysis was carried out using HyperDoppler software. The authors obtained, in general, the expected results (in patients, some ECHOCG parameters (EF, LV myocardial mass, parameters of diastolic function) were worse), but not the vortex analysis data.

In addition, when reviewing, I had questions and comments to which I would like to receive answers from the authors of the article.

  1. The title and purpose of the article does not match the content (it is not indicated that individuals were evaluated during physical training programs).

Thank very much  for  this suggestion :  the  title  has been  modified  containing this  aspect .The aim  has been added in the  abstract.

  1. Characteristics of the surveyed is presented insufficiently. For example, age is listed as 55+/- without DM data. In addition, it is necessary to compare data in groups (gender, age, BMI, presence of concomitant diseases, received therapy, heart rate, SBP and DBP). These parameters can have a significant impact on the ECHOCG parameters and on the revealed differences between the groups.

The missed  data of the  age have been added . No differences  have been  found for the  parameters  considered particularly within the transplanted group.  

 The text has been updated

  1. I think table 1 is redundant, if necessary, this information can be presented as an additional column in table 2. However, given the different nature of the groups (patients after transplantation and healthy controls), this is hardly appropriate.

 We appreciate this  suggestion . We  think , however it  is  hard  to avoid  to  collect  all the  data  in the  text  or to add in a separate  column in the  tab2 . This tab show a  comparison  of the  two groups , on the contrary  the tab 1 collect all the  data  of the  subjects .

  1. It is not clear how loads were selected in patients receiving beta-blockers and how many such patients were in the study?

None of them received beta- blockers as reported in the   paper (calcium channel blockers, alpha-blockers) . We  have specified  in the text

  1. In the group after transplantation, there is no information about the terms after the operation and the main biochemical markers.

 Thank  you  for this  suggestions : all  the biochemical parameters  were normal . In fact  the subjects are normally screened in the  ambulatorial setting before  to be sent to  the  sports  medicine  center .  The time from transplantation was at least 1 year . The  range was 10 years .

  1. Reading lines 195-197 ("The range of the vortex parameters has been in investigators and DCM in a study demonstrating significant differences among them, suggesting a potential role in the definition of the single group investigators"), one gets the impression that source link missing.

The reference has been added  n 15

  1. The Discussion section is poorly written - there is practically no comparison of the obtained data with previously published works, it should be revised.

 The discussion has been  entirely re-written

  1. I question the authors' assertion at lines 209-211 ("In addition, it could be helpful to use the HyperDoppler analysis to early detect potential damage in the diastolic time before having a systolic impairment") because differences in diastolic function scores between groups were , but not according to HyperDoppler analysis.

We are in agreement with you .  the sentence of the paper  is not correct. We hypothesized that the correlation found between  two vortex paramters ( Vorticity Fluctuation  and  E/A and   septal E1)  and  vortex area with A1 septal value , confirm  the  potential  role  of the vortex  parameters with diastolic phase . It could be reasonable to hypothesized  that  in presence of higher the  diastolic dysfunction expressed by E/E1  higher ratio value, these parameters show a peculiar trend  characterized by a reduced vortex area  and  higher vorticity  fluctuation  as reported in tab 3 despite at present   not significant. We  have modified the text 

  1. The numbering of subsections of the manuscript is somewhat chaotic, it needs to be clarified.

 The numbering of the subsections has been  corrected.

  1. The article does not contain a Conclusion section.

The conclusion have been added

Reviewer 2 Report

Abstract:

Abbreviations such as GLS, LV, IVSD has to be defined at their first mention

Please consider rewriting the abstract in a clearer and precise scientific language

In line 20: Authors mentioned 47: 33 transplanted and 17 healthy subjects. While in the text the number of healthy subjects were 14?

Method:

Your experimental group were patients with history of transplantation (liver and kidney), on the other hand, the control group are healthy subjects. How author can justify comparing healthy subjects with patients after transplantation?  

The study groups should be all patients after transplantation, with exercise training and without training then compare the clinical outcomes!

In addition, patients with liver transplantation are different from patients with kidney transplantation. Authors should not mix them in one group they should be separated and then compared!

English language is required to be improved throughout the manuscript 

Author Response

Abstract:

Abbreviations such as GLS, LV, IVSD has to be defined at their first mention

 The  modifications requested  have been  made

Please consider rewriting the abstract in a clearer and precise scientific language:

Abstract has been modified

In line 20: Authors mentioned 47: 33 transplanted and 17 healthy subjects. While in the text the number of healthy subjects were 14?

 The correction has been made . Thank  for notifying

Method:

Your experimental group were patients with history of transplantation (liver and kidney), on the other hand, the control group are healthy subjects. How author can justify comparing healthy subjects with patients after transplantation?  

The comparison  of a solid organs transplanted subjects  with  control  healthy  and  active subjects  offer us  the possibility  to compare  the  range of the parameters found  and any eventual correlation with systolic or diastolic standar  parameters . Consider that the literature regarding the vortex parameters  is  at the  initial phase of investigation and in addition, no data are available  about the specific population with  exclusion of Prof D. Mele paper (Noninvasive Evaluation of Intraventricular Flow Dynamics by the HyperDoppler Technique: First Application to Normal Subjects, Athletes, and Patients with Heart-J. Clin. Med. 2022, 11, 2216. https://doi.org/10.3390/jcm11082216) . This  is  the  first investigation in post transplantation .

The study groups should be all patients after transplantation, with exercise training and without training then compare the clinical outcomes!

 This  is a great  idea that will represent the future investigation . Our laboratory  is intensily dedicated to this follow  up –We have planned this protocol for  the  next  investigation. We  apologized is not yet avaliable .

In addition, patients with liver transplantation are different from patients with kidney transplantation. Authors should not mix them in one group they should be separated and then compared!

 Thank  you  for  giving  the  opportunity  to clarify  this  important aspect . Despite we are in agreement , however  we respectfully underline that they  are all  solid  organs transplanted subjects  and  in this  preliminary approach  to this  investigation , the study wants  to represents  this  aspect . We have added a specific comment in the conclusion session . In addition the physical  activity level and the associated cardiovascular disease ( hypertension and diabete) are not different in both.

Round 2

Reviewer 1 Report

The authors significantly corrected the text of the manuscript and answered all my comments and questions. I have no other comments.

No comments

Reviewer 2 Report

No further comments